# Critical Assessment of *Streptomyces* spp. Able to Control Toxigenic Fusaria in Cereals: A Literature and Patent Review

**DOI:** 10.3390/ijms20246119

**Published:** 2019-12-04

**Authors:** Elena Maria Colombo, Andrea Kunova, Paolo Cortesi, Marco Saracchi, Matias Pasquali

**Affiliations:** Department of Food, Environmental and Nutritional Sciences, University of Milan, 20133 Milan, Italy; Elena.colombo25@gmail.com (E.M.C.); andrea.kunova@unimi.it (A.K.); paolo.cortesi@unimi.it (P.C.); matias.pasquali@unimi.it (M.P.)

**Keywords:** mycotoxin, deoxynivalenol, fumonisin, biocontrol, antagonism, bioactive compounds, wheat

## Abstract

Mycotoxins produced by *Fusarium* species on cereals represent a major concern for food safety worldwide. Fusarium toxins that are currently under regulation for their content in food include trichothecenes, fumonisins, and zearalenone. Biological control of *Fusarium* spp. has been widely explored with the aim of limiting disease occurrence, but few efforts have focused so far on limiting toxin accumulation in grains. The bacterial genus *Streptomyces* is responsible for the production of numerous drug molecules and represents a huge resource for the discovery of new molecules. *Streptomyces* spp. are also efficient plant colonizers and able to employ different mechanisms of control against toxigenic fungi on cereals. This review describes the outcomes of research using *Streptomyces* strains and/or their derived molecules to limit toxin production and/or contamination of *Fusarium* species in cereals. Both the scientific and patent literature were analyzed, starting from the year 2000, and we highlight promising results as well as the current pitfalls and limitations of this approach.

## 1. Introduction

Mycotoxins are extracellular metabolites produced by filamentous fungi that contaminate cereals, grains, fruits, and vegetables. The most important *Fusarium* toxins are trichothecenes, zearalenone (ZEN), and fumonisins (FBs), that are dangerous for human and animal health, and their presence in food is regulated worldwide [1]. Mycotoxin co-occurrence in food is a real and relatively underestimated issue [2], as is the modification of toxins by plant metabolism (creating masked mycotoxins) [3]. Both factors mean that the levels of toxins measured in food, and therefore being ingested, are significantly underestimated. Due to this, it is likely that normative limits will be lowered by the regulatory agencies in the future.

Cereals, the staple foods of diets all over the world, are perfect hosts for pathogenic and toxigenic fungi and represent one of the main sources of mycotoxin contamination for humans and animals [4]. Among toxigenic species, *Fusarium* spp. (Division Ascomycota) are major producers of mycotoxins in cereals [5].

Trichothecenes A and B are mainly associated with Fusarium head blight (FHB) and crown rot (FCR) in wheat and barley. The major group of *Fusarium* spp. responsible for these diseases includes *Fusarium graminearum* species complex (FGSC; [6]) that exhibits a diverse distribution of species across the different continents [7]. The most important species are *F. graminearum, F. culmorum*, and *F. pseudograminearum* [8,9]. Grain quality decrease and yield are of concern [10]. The trichothecenes type B are the most prevalent and comprise deoxynivalenol (DON) and nivalenol (NIV) and their acetylated forms 3-ADON, 15-ADON, and 4-ANIV [11]. They are immunosuppressant and neurotoxic and cause intestinal irritation, leading to feed refusal in livestock [12,13]. In maize, *F. graminearum* and other related species were found to be associated with Fusarium ear rot (FER), contaminating grains with ZEN. ZEN displays estrogenic activity, causing reproductive problems in animals, in addition to cytotoxic and immunosuppressive effects [14,15].

Ear rot in maize is also caused by *F. verticillioides* (syn. *F. moniliforme* [16]) and *F. proliferatum*, which produce fumonisins [17]. Fumonisins have been classified as Group 2B carcinogens (i.e., as possibly carcinogenic to humans [18]), and fumonisin B1 (FB_1_) is the most abundant analogue found in contaminated samples [19]. Moreover, *Fusarium* spp. infecting cereals can also produce other minor mycotoxins with cytotoxic effects such as enniatins, beauvericin, and moniliformin. Knowledge gaps regarding the occurrence, toxicity, and toxicokinetic data for these compounds in cereal crops represent a major and immediate problem [20].

*Fusarium* spp. infections of cereals are therefore a major concern for both the growers and the food chains associated with the processing of grains. Several control strategies against this complex group of pathogens have been developed and include host resistance, the application of fungicides, and the implementation of specific agricultural practices [21]. However, effective management of *Fusarium* pathogens and the related toxins cannot be achieved through the use of a single control strategy because each has its own limitations [22]. Therefore, at least in Europe, integrated disease management is urgently needed, favored by European Regulation 1107/2009/EC and European Directive 128/2009/EC [23,24]. Moreover, biocontrol approaches are becoming increasingly important due to the limitation on the use of certain fungicides. Among the biocontrol agents (BCAs) used to control toxigenic *Fusarium* spp. in cereals, bacteria have shown a number of successful outcomes. For instance, strains of *Bacillus* spp. [25,26,27,28], *Brevibacillus* sp. [29], *Pseudomonas* spp. [27,30], and *Lysobacter enzymogenes* [31] were applied to limit pathogen development, reducing disease severity and mycotoxin production. Microbial communities or single strains were also tested to detoxify contaminated substrates as reviewed by McCormick in 2013 [32].

Bacteria of the genus *Streptomyces* display promising plant growth-promoting features and biocontrol efficacy against plant pathogens. They belong to the phylum of Gram-positive Actinobacteria, which is one of the largest taxonomic units within the bacterial domain, and include microorganisms relevant to human and veterinary medicine, biotechnology as well as ecology [33]. Streptomycetes are the most abundant actinobacteria in soil [34]. They display a unique life cycle, and after germination grow through a combination of tip extension and the branching of hyphae. They first form a vegetative mycelium firmly attached to the growth substrate and, subsequently, due to nutrient depletion and under environmental stress signals, develop an aerial mycelium. Each aerial hypha then differentiates into a long chain of pre-spore compartments which subsequently mature into individual spores [35]. The ability to produce a variety of secondary metabolites, including anti-infective agents, has an important ecological role including the inhibition of competitors during the transition from mycelial to aerial growth [36]. These various characteristics enable them to colonize different substrates and establish symbiotic interactions with plant tissues and other eukaryotes [37]. The ability to produce numerous secondary metabolites means they are the most exploited bacterial genus in natural product research. Notably, more than half of all antibiotics in current clinical use are derived from actinobacterial secondary metabolites [38]. Furthermore, *Streptomyces* spp. were evaluated as plant growth-promoting bacteria (PGPB), as they can inhibit pathogen development, enhance nutrient uptake by mineral solubilization, and increase plant growth by nitrogen fixation and phytohormone synthesis [39]. Streptomycetes have therefore been investigated for their possible use in agriculture, including cereal crops [40].

The diversity of secondary metabolite production plus their reported endophytic features make the genus *Streptomyces* a perfect candidate to control toxigenic *Fusarium* spp. development and related toxin production [41,42]. Endophytic microorganisms have been reported as useful antagonists against Fusarium head blight, and are able to reduce disease severity on spikelets [43]. Nevertheless, the incredible diversity and potentiality of these microorganisms against mycotoxigenic fungi and their possible influence on toxin accumulation have been rarely explored and deserve further investigation [44]. This review describes reports in which streptomycetes, or molecules derived from them, were exploited against *Fusarium* spp., and will pay special attention to the possible influence on toxin production. The scientific and patent literature were analyzed from the period 2000–2018.

## 2. Critical Assessment of Literature

Despite the huge amount of literature regarding the biological control of *Fusarium* mycotoxigenic isolates in cereals, only two products have found a consistent market niche [45]. These are based on *Pseudomonas chlororaphis* and *Pythium oligandrum* and are marketed in Europe as Cerall^®^ (Belchim Crop Protection) and Polyversum^®^ (Biopreparáty/De Sangosse), respectively [46]. Furthermore, no *Streptomyces* product is officially registered to be used for this purpose [47]. The main obstacles for biocontrol agents are due to the lack of consistency when microbial inoculants are applied under complex environmental conditions, and to the complexity of finding appropriate formulation and timing for application [48]. Biological, ecological, toxicological, and regulatory cost factors also influence the effectiveness and marketability of biological control products [49].

In order to verify the status of research using *Streptomyces* strains, and their derived molecules, to limit toxigenic *Fusarium* spp. infections and/or toxin contamination, we screened the published literature. To critically assess the status of each research paper, a set of definitions describing the type of study and their accuracy was established as follows.
*Streptomyces* species definition. Species identification is essential as approximately 10 *Streptomyces* species have been described as plant pathogens, causing economically important diseases on underground plant structures such as tuber/root crops. The best studied and characterized of these is *Streptomyces scabies* which causes potato scab [50,51]. Moreover, from a food safety perspective, *Streptomyces* isolates are able to produce dangerous metabolites for human and animal health, such as antimycin A found on wheat and barley grains [52]. Therefore, it is essential that species and strain characterization is performed accurately.In vitro testing for antifungal activity. This is generally the first step for identifying antifungal microorganisms or molecules produced by them. Such studies help define the mechanism(s) of action of the *Streptomyces* species(s) and lead to the identification of potential interactions with the target organism. Assessment of bioactivity should consider the diversity of targets (verifying if pathogen diversity influences the consistency of the BCA or derived product). Indeed, specific interactions occur among bacterial and fungal strains [53] and this may impact the biocontrol capability of a strain [54,55].The effect of culture media in the bioassays in vitro. Media composition modulates secondary metabolite production in actinomycetes [55,56,57], and optimizing laboratory selection procedures should broaden the number of interesting BCAs that can be identified.The use of fermentation extracts to perform bioassays. During screening procedures, it would be ideal to identify the metabolite(s) responsible for the observed antifungal effect. The screening of crude extracts is generally followed by further steps of purification and chemical analysis, and retesting of purified compound(s) [58].Evaluation of the antifungal mode of action. Risks concerning the use of these antibiotic-producing bacteria associated with events of horizontal gene transfer and the development of antibiotic resistance are still under debate within the scientific community [44]. However, given the current legislative requests [59], understanding the mode of action is essential in order to proceed with the registration of a BCA, in order to avoid risks of spreading dangerous metabolites for human and animal health in the environment [60].Assessment of the ability to colonize treated plant organs. Many BCAs are rhizosphere-colonizing microorganisms and can be applied as seed coatings [61]. However, some *Streptomyces* spp. can exhibit endophytic behavior, colonizing different parts of the plant (e.g., roots, stem, leaves) [41]. Some BCAs exhibit activity both in the rhizosphere and after infection of the plant and function inside the root at the same time. Therefore, the colonization niche of the strain should be investigated in order to warrant a consistent protection [62]. These studies are fundamental to providing an assessment of the durability of the protection warranted by the BCA.Testing the influence of complex environmental conditions. As for pathogens during disease development, antagonist strains are influenced by environmental factors that strongly impact the ability of the BCAs to exert their biocontrol activity [40]. Assessing the impact of environmental parameters on BCAs using both greenhouse and field trials is essential to selecting strains with consistent biocontrol activity.Assessment of antifungal and plant growth-promoting effect in planta. This step is essential, given that the BCA will ultimately be employed in the field. Very often, there is poor correlation between in vitro and in planta trials [55,63,64]. Moreover, the wide range of metabolites produced may have direct influences on plant development, altering growth and plant fitness both positively and negatively [39]. Indeed, negative effects cannot be underestimated—some *Streptomyces* can be pathogens (see before) or produce phytotoxic and herbicidal substances [65].Assessment of the method used for application. Selecting an appropriate delivery system for the BCA as well as an optimized formulation can determine its efficacy in the field [66].The effects of the BCA on the pathogen inoculum in planta. Due to the complex epidemiology of *Fusarium* diseases in cereals, quantification of the pathogen in planta is important to verify if the treatment can, for example, effectively reduce the source of overwintering inoculum, limiting the infection pressure at the subsequent infection season [67].Quantification of the mycotoxin. It is essential to verify if the BCA limits toxin production, specifically given that there is a lack of full correlation between the presence of the fungus and the amount of toxin that is found in the grains [68,69]. Moreover, some secondary metabolites can limit toxin production without impairing growth of the pathogen [70]. Biological interactions can also lead to unexpected crosstalk between the BCA and pathogen that can lead to an overproduction of toxins and secondary metabolites [71,72,73,74,75].

## 3. Literature Analysis

To guide future implementation of biocontrol research using *Streptomyces* spp., it is essential to identify the strengths and weaknesses of past and present research in this domain. Therefore, we reviewed the published literature, focusing on the methods used for the selection of promising biocontrol streptomycetes and on the results achieved.

We searched the Scopus and Google Scholar databases for articles including the words “*Fusarium*” and “*Streptomyces*” that were published during the timeframe 2000–2018. The resulting articles were read and individually screened leading to the identification of 63 articles that dealt with the ability of *Streptomyces* or their secondary metabolites to limit the growth or toxin production of toxigenic *Fusarium* spp. in cereals (Table 1).

*Streptomyces* spp. or their derived molecules have been tested mostly against *Fusarium* spp. producing trichothecenes, including DON. The species investigated are all usually found to infect cereals and include *F. graminearum*, *F. culmorum*, *F. poae*, *F. cerealis, F. sporotrichioides*, and *F. equiseti.* The most studied interactions address the wheat–*F. graminearum* pathosystem, which is the most important cause of DON (and derivatives) accumulation in grains [76]. Less frequently, streptomycetes have been tested against fumonisin producers in maize, all belonging to the *F. fujikuroi* species complex [77].

### 3.1. *Streptomyces* Identification

Regarding the identification of *Streptomyces* species, most studies focused on the integration of morphological and molecular characteristics. Given the complexity of streptomycete biology [33], the use of 16S rRNA alone as molecular marker is not sufficient to achieve species discrimination. Multi-locus sequence typing [78] integrated with biochemical and morphological identification would be a preferred option, but none of the studies used this approach. On this basis, all the species identifications reported in the selected papers should be treated with caution. Looking forward, the increasing number of *Streptomyces* strain genomes now available may help in correct species identification [79].

### 3.2. Screening for Antifungal Activity: In Vitro Tests

Among the selected articles, in vitro testing is the most commonly used first-line screening method. Indeed, dual-culture assays on solid media are exploited in all the studies as a preliminary screen, evaluating the inhibition halo between the growth of the streptomycete and the fungal target or measuring the radial growth of the *Fusarium* colony in comparison with an untreated control to obtain a percentage of growth inhibition. Rather than use these standard in vitro inhibition assays, some research groups [80,81,82] characterize the type of interactions occurring in dual culture by using the index of dominance (ID) [83]. The ID consists of visually observing antagonist and pathogen growth in dual culture, testing different media or water activity (*a_w_*) of the culture medium, and classifying the type of interactions occurring based on predefined scores, namely, mutual intermingling (1/1), mutual inhibition on contact (2/2), mutual inhibition at a distance (3/3), dominance of one species on contact (4/0), and dominance at a distance (5/0). This method evaluates whether the inhibition is due to the production of antifungal metabolites diffusible in the media or whether the mycelium is parasitized by the antagonists. Moreover, the negative effect of the target pathogen on the potential antagonists can be noted. Therefore, the selection of biocontrol agents is carried out by evaluating the biocontrol interactions (e.g., mycoparasitism, competition, or antibiosis) established under different growth conditions.

For most reports, growth of the *Streptomyces* inoculum to some predefined point usually takes place on agar media before addition of the pathogen in order to allow a complete establishment of these growing bacteria [80,84].

The use of a diverse range of growth media and fungal strains was evaluated in our analysis, given the importance that these criteria have in the estimation of the biocontrol activity in vitro [55]. Interestingly, the influence of growth media was seldomly evaluated in these types of experiments [84,85] as well as the assessment of antifungal activity on different *Fusarium* strains belonging to a single species [80,86,87,88].

Given the lack of a standardized protocol when performing dual-culture assays (e.g., *Fusarium* strains on which the biocontrol activity should be tested, position and distance between streptomycetes and *Fusarium* strain inoculum, timing of observation after pathogen inoculum, culture medium), it is difficult to compare the results between studies. However, here we report some examples of the wide range of activities recorded against mycelial proliferation. For instance, growth inhibition percentages against *F. graminearum* and *F. verticillioides* ranged from the weakest (<20%) [84,89] up to 60–90% of inhibition [87,90]. Yekkour et al. [91] obtained different levels of inhibition in dual culture for isolated streptomycetes—indeed, only 6 out of 133 isolates displayed an anti-*Fusarium* activity and in particular only *F. culmorum* was significantly inhibited (inhibition halo >20 mm). Less sensitive fungal species were *F. moniliforme*, *F. sporotrichoides*, *F. graminearum*, and *F. proliferatum* [91].

### 3.3. Evaluation of Antifungal Mechanism of Action

The importance of the identification of any antifungal molecules involved in the bioactivity led some researchers to achieve a complete characterization of the compounds involved. The fermentation process and the optimization of all the parameters (e.g., medium, agitation rate, pH, temperature) were strain- and laboratory-dependent [92,93,94,95]. For instance, it has been reported that some of the *Streptomyces* strains which are active against *F. moniliforme* on solid media lack antibiotic production in submerged liquid culture, highlighting the importance of an appropriate optimization of laboratory procedures and media in the stimulation of secondary metabolites [96]. The first attempt of compound purification is commonly carried out by crude extract fractionation [97,98]. Often the bioactivity of the selected strain is not related to a single mechanism, and different metabolites, enzymes, or volatile organic compounds likely contribute to the overall antifungal activity. Many studies exploited the fermentation broth as a source of bioactive compounds [99,100,101]. Therefore, several compounds were purified and tested against toxigenic *Fusarium* spp. For example, strain PAL114 produced saquayamycins A and C which inhibited the growth of *F. culmorum* at the minimum inhibitory concentrations of 75 ug/mL [102]. Three allelochemicals (5,7-dihydroxyflavone, 5-hydroxy-7-methoxyflavone, and di-(2-ethylhexyl) phthalate) able to inhibit mycelial growth of *F. graminearum* were isolated and purified from the fermentation broth of *Streptomyces* sp. 6803 [103]. In vitro cultures of *Streptomyces* sp. 201 produced 2-methylheptyl isonicotinate able to inhibit the growth of *F. moniliforme* more efficiently than a natural analogue (isoniazid) [98]. On the other hand, modest activity was observed by the metabolites extracted from *Streptomyces* LZ35 against *F. verticillioides* [104]. For several studies, chitinase activity, rather than antibiotic production, was shown to play a role in the antifungal mechanism [105,106,107,108]. In addition, new antifungal proteins have been characterized, such as the one isolated from *Streptomyces* sp. C/33-6 culture supernatants which displayed a fungicidal activity, determining complete inhibition of conidia germination of *F. graminearum* [109].

Secondary metabolites exhibiting anti-*Fusarium* activity can also include volatile organic compounds (VOCs). For example, *Streptomyces alboflavus* TD-1 was able to reduce the mycelial growth of *F. moniliforme* when volatile metabolites were applied as fumigants [110]. Inhibition of growth, sporulation, and conidial germination has been recorded when culturing this strain on wheat seeds. In addition, the VOC activity increased the fungal membrane permeability as observed by significant leakage of mycelial materials. Chemical analysis of these VOCs identified a high quantity of 2-methylisoborneol and 2-methyl disulphide, which were further tested for their antifungal activity [110,111]. VOC production was also linked to the antagonist activity of *Streptomyces philanthi* RM-1-138 cultured on wheat seeds, which inhibited mycelium growth of *F. fujikuroi* by 50% [112]. Chemical analysis showed that a complex mixture of volatile metabolites was involved [112].

It is evident from our analysis that the biocontrol activity of *Streptomyces* strains involves a large range of bioactive molecules. The exploitation of *Streptomyces* spp. has been, and will in future also be, hindered by the variability of the production of these metabolites. Therefore, to exploit the huge diversity of streptomycetes for successful disease management, different factors, such as the age of the fungal colony, culture conditions, temperature, and other environmental parameters, will have to be carefully studied, even at the very early stages of investigation. Transferring the outputs of these laboratory studies to the field remains one of the major challenges in exploiting *Streptomyces* spp. as BCAs for tackling toxigenic *Fusarium* spp.

### 3.4. Assessment of Streptomycete Effects in Planta

The literature reports a lack of durable and consistent effects when streptomycetes or commercially available formulations have been applied in greenhouse experiments and field trials [40]. It is likely that the ability to cope in a complex environment, which comprises the plant, the presence of the pathogens as well as several abiotic factors, varies depending on the fitness of the strain and its formulation in the field. For this reason, verifying the level of colonization achieved by the strain when used as BCA is essential to confirming its ecological fitness. Only a few papers have addressed this question in detail. It is notable that most of these were published recently, which indicates an increasing level of attention regarding *Fusarium*–plant–*Streptomyces* interactions [113,114].

Moreover, in planta experiments are essential during the process of BCA selection to confirm their ability to significantly decrease *Fusarium* spp. infections. Indeed, BCAs can influence crop growth and disease severity, and can reduce *Fusarium* inoculum levels on stubble after harvest as well as ideally the presence of mycotoxins [115]. However, only a limited number of studies (*N* = 16) performed complete in planta studies. The application of streptomycetes was tested on seeds [84,116,117,118], on the main emerged spike [80,84,86,87] as well as wheat stubble [86]. Indeed, these bacteria can contribute to the reduction of FHB on wheat at different times in the *Fusarium* spp. life cycle. In a research study conducted by Palazzini et al. [80] in 2007, isolates from wheat anthers were applied to wheat heads grown in greenhouse and, after 16 days, their influence on FHB severity was estimated. Despite the slight reduction of disease symptoms in comparison to the control, streptomycete BRC 87B decreased the DON content in spikes to below a detectable level. For this reason, in a subsequent study it was tested in the field, showing the ability to decrease FHB severity and DON amounts, as well as the *F. graminearum* inoculum on wheat stubble [86].

Testing the efficacy in the field also requires specific assessments of the way the strains are inoculated. For example, the use of a Korean strain isolated from rice kernels led to a significant reduction of the disease severity after its inoculation using a spore spraying method that was not achieved using the point inoculation method on wheat heads [84]. This is actually the only study where the influence of the BCA application method was taken into account, and shows that, depending on the application of the BCA, different results can be obtained [47].

Differences in the level crop protection have also been reported against other *Fusarium* spp. For instance, two *Streptomyces* strains designated as DAUFPE 11470 and DAUFPE 14632 were isolated from maize rhizosphere in Brazil and tested against maize seed pathogenic fungi. Treatments on seeds with biomass derived from streptomycete fermentation or with cell-free filtrate reduced significantly *Fusarium subglutinans* incidence on stored maize seeds [119]. The same strains were also tested as spore suspension to assess their effects on seedling blight caused by *F. moniliforme* in greenhouse [120]. Bacterial treatments significantly reduced disease incidence compared with the controls, with protection level variable according to the tested pathogen inoculum concentrations. Indeed, the disease incidence was significantly reduced at low and high antagonist and pathogen concentrations, respectively. Moreover, their ability to reduce chlamydospore germination was assessed—the percentage of germinated propagules was evaluated after antagonist treatments in sterilized soil added with glucose, to recreate the natural environment and enhance spore germination. The addition of glucose increased propagule germination in all the treatments, but the presence of the antagonists decreased this parameter up to 65%. This study stressed therefore the important influence of both antagonist and pathogen concentrations and the presence of nutrients in the final biocontrol efficacy obtained in planta [120].

*Streptomyces* strains, as reported above, can be helpful in reducing disease symptoms, acting also as plant growth-promoting bacteria. Despite the wide range of metabolites produced by them, their ability to influence plant development has been seldomly studied by the current literature addressing the biological control properties of the strain. A few positive examples include the report of a negative influence on seed germination and seedling development [91] as well as an improvement in plant growth parameters [87].

### 3.5. Evaluation of Streptomycete Activity against Mycotoxin Production

As noted above, it is essential to accurately determine the concentration of mycotoxins present in grains destined for human or animal consumption. Similarly, verification of the toxin content under experimental conditions is vital for the future of potential streptomycete biocontrol agents. Indeed, it should be possible that the reduction of disease severity does not positively correlate with a reduction of the mycotoxin content in grain samples. So far, only one research group has evaluated the reduction of DON mycotoxins by *Streptomyces* strains isolated from wheat anthers, in comparison to the level of infection, in vitro, in greenhouse, and in the field [80,86]. Indeed, they showed that their streptomycete strains (BRC 87B and BRC 273) were able to significantly reduce DON levels on wheat grains, without influencing disease severity caused by *Fusarium* infections [80]. This suggests the existence of a specific mechanism of inhibition uncoupling fungal fitness and toxin production. Follow-up research by the same group evaluated in the field the use of BRC 87B, which showed strong inhibition of DON production in wheat spikes [86].

Preliminary in vitro studies have also been conducted to verify the ability of streptomycetes to limit fumonisin accumulation. Strains isolated from soil samples amended with different organic manures by Nguyen et al. were tested against fumonisins FB_1_ and FB_2_ production by *F. verticillioides* [121]. They significantly decreased (by up to 98.2%) the level of FB_1_ and FB_2_ in agar plate cultures [121]. Inhibition of FB_1_ accumulation on milled maize agar was also demonstrated in another in vitro study using *Streptomyces* sp. AS1 [122], a strain isolated from peanuts in Egypt. Further, El-Naggar et al. [123] showed the ability of *Streptomyces* isolates to reduce accumulation of a wide range of mycotoxins including total aflatoxins, fumonisin, zearalenone, T-2 toxin, AOH, and AME. However, the identity of the *Fusarium* spp. producers was based only on morphological characteristics and should be considered with caution.

## 4. Patent Search

To have a complete overview of the work using *Streptomyces* against toxigenic fusaria, a research of the major patent databases was carried out. Using both Espacenet and Orbit intelligence, a total of 233 results were obtained using the keywords “*Fusarium*” and “*Streptomyces*”. By manually screening the titles and abstracts, a total of 25 patents were retained and added to Table 2. Given the use of different languages (most not English), only certain abstracts could be accessed, and it was therefore not possible to apply the same critical criteria used in our literature search. Most of the patents claimed general activity of strains and derived molecules against a large set of microorganisms including toxigenic fusaria. Only a single patent in its claim directly addressed the ability to limit *F. graminearum* growth on cereals [147]. Two documents patented the antifungal metabolites isolated from streptomycete strains and tested them against toxigenic *Fusarium* spp. [148,149]. The other patents are related to specific formulation methods, using live streptomycetes, proposed as biocontrol products against plant pathogens, among them *Fusarium* spp. of cereal crops.

Interestingly, most of the patents are concentrated in the last five years (Table 2), therefore further developments could also be expected towards novel industrial applications in the near future.

## 5. Conclusions and Perspectives

Our review of the literature and patents clearly identifies a growing interest in the use of *Streptomyces* spp. as biological control agents against toxigenic *Fusarium* spp., both to inhibit growth and to limit toxin accumulation (contamination). However, it is clear that for the majority of the available studies, the findings are preliminary. In most cases, a clear understanding of the role of the BCA, the identification of the molecules or mechanisms of inhibition, as well as the fungal targets are lacking [172]. Moreover, most of the data are limited to laboratory in vitro experiments and lack validation in planta or in the field.

The future of research on streptomycetes as biocontrol agents for *Fusarium* will need to integrate diverse expertise and may profit from new methods able to better mimic in the laboratory interactions occurring in the field [55]. Novel formulation and application techniques will be needed to enable individual beneficial microbes and microbial consortia to exert their activity in a consistent manner for different crops and soils [173]. For instance, one biocontrol approach to further investigate could be combining multiple strains to build consortia able to exert complementary activities [174]. Indeed, understanding the ecological role, including specific interactions with other microorganisms and the host, is essential for developing effective and long-lasting approaches of biocontrol. Reaching a better understanding of microbes–*Fusarium* interactions could help to provide effective biocontrol strains among natural endophytes present in the wheat microbiome [175] and within graminaceous plant rhizosphere [176]. The effect of specific interactions as well as the ability to shift metabolic profiles within the same *Streptomyces* species, niche and also among individuals [177], suggest that studies on the efficacy of strains should encompass a broad range of conditions mimicking the agricultural milieu [55]. Appropriate fitness tests able to predict the behavior in the field are needed at the selection level. Novel BCAs or their metabolites could also be identified and produced integrating appropriate novel genome editing [178] as well as adaptive evolution techniques [179]. A better understanding of secondary metabolite regulation during the interaction with fungi will help to increase their discovery for agricultural purposes [53].

Our analysis of the literature leads to the observation that each single paper only addresses a few aspects of the proposed criteria that would have to be evaluated in identifying effective *Streptomyces*-based BCAs. This review may serve as a proposal for future research efforts which will likely profit from an integrated analysis of the different parameters that we have identified.

The increasing interest within industry, proven by the increasing number of patents that address and refer to the use of *Streptomyces* spp. to limit *Fusarium* spp. in grains, is a further indication of the potential role that this powerful group of microorganisms can play in the future of agricultural research. In conclusion, by performing a complete analysis of the literature regarding the use of *Streptomyces* spp. for the biological control of mycotoxigenic fusaria, we identified a set of parameters that we consider essential for enabling their implementation for biological and toxin contamination control. Our review suggests that streptomycetes have the potential to play a crucial role both as BCAs, and as producers of novel inhibitory molecules, for the combined control of *Fusarium* infection and to limit the accumulation of mycotoxins in crops [180,181].

## Figures and Tables

**Table 1 ijms-20-06119-t001:** Published studies regarding the efficacy of *Streptomyces* spp. (and derived molecules) against *Fusarium* toxigenic species in vitro, in planta, and under different environmental conditions. The methods used for the identification of the *Streptomyces* strain are also reported. Data were obtained combining the results of Scopus and Google Scholar searches with the following search words, “*Fusarium*” and *“Streptomyces*”, limiting the period of publication from 2000 to 2018. Legend: M (Morphological identification), B (Biochemical identification), BCA/s (Biocontrol agent/s), GC (Growth chamber), G (Greenhouse), F (Field), * possibly misleading identification.

*Fusarium* spp. Studied	Streptomycete Identification	In Vitro Tests for Antifungal Activity	Influence of Pathogen Diversity	Influence of Culture Media on BCAs	In Vitro Tests Using BCA Extracts	Evaluation of Antifungal Mode of Action	BCAs’ Survival on Plants	Environment of Trials in Planta	Evaluation of BCA Application	BCAs’ Effects on Plants	BCAs’ Effects on Disease	BCAs’ Effects on Fusaria Inoculum	Toxin Measurement	References
*F. avenaceum*		x			x	x								[95]
*F. avenaceum*										x	x			[124]
*F. avenaceum*, *F. oxysporum*, *F. solani*					x	x								[125]
*F. coeruleum*; *Gibberella saubinetii*	16S rRNA	x			x	x								[101]
*F. crookwellense*; *F. oxysporum*		x				x								[107]
*F. culmorum*		x			x	x								[102]
*F. culmorum*	M/B/16S rRNA	x												[126]
*F. culmorum*		x			x	x								[99]
*F. culmorum*	M	x				x		GC		x	x			[116]
*F. culmorum*						x	x	GC		x	x			[114]
*F. culmorum*, *F. moniliforme*, *F. sporotrichoides*, *F. graminearum*, *F. proliferatum*	16S rRNA	x						GC		x	x			[91]
*F. culmorum, F. equiseti, F. proliferatum, F. graminearum, F. sporotrichioides, F. moniliforme, F. oxysporum*	M/16S rRNA	x			x	x								[127]
*F. culmorum, F. graminearum, F. proliferatum, F. oxysporum*	M/B/16S rRNA	x				x								[128]
*F. culmorum, F. oxysporum*	16S rRNA	x			x	x								[108]
*F. culmorum, F. oxysporum*		x		x	x	x		GC		x				[129]
*F. culmorum; F. graminearum; F. oxysporum*								G			x			[118]
*F. fujikuroi*		x			x	x								[112]
*F. graminearum*	M/B/16S rRNA	x			x	x		G		x	x			[130]
*F. graminearum*		x				x								[106]
*F. graminearum*		x			x	x				x				[103]
*F. graminearum*	M/B/16S rRNA	x			x	x	x			x				[113]
*F. graminearum*	M/16S rRNA	x		x				G	x	x	x			[84]
*F. graminearum*	M/B/16S rRNA	x			x	x								[131]
*F. graminearum*	M/B/16S rRNA	x	x					G			x		x	[80]
*F. graminearum*			x					F			x	x	x	[86]
*F. graminearum*			x					F		x	x	x	x	[88]
*F. graminearum*		x		x	x	x								[85]
*F. graminearum*								G; F				x		[132]
*F. graminearum*		x	x			x		G		x	x			[87]
*F. graminearum F. culmorum*	16S rRNA	x												[43]
*F. graminearum, F. culmorum, F. oxysporum*	M/B	x				x							x	[133]
*F. graminearum, F. oxysporum*		x			x	x				x				[134]
*F. graminearum, F. oxysporum*		x			x	x								[135]
*F. graminearum, F. oxysporum, F. solani*		x			x	x								[105]
*F. graminearum, F. proliferatum, F. sporotrichioides, F. oxysporum*		x			x	x								[109]
*F. graminearum, F. verticillioides, F. culmorum*	M/ 16S rRNA	x						G		x	x			[117]
*F. graminearum, moniliforme, F. oxysporum, F. solani*	M/B/16S rRNA	x			x	x								[100]
*F. graminearum; F. moniliforme*	M/B/16S rRNA	x		x	x	x								[136]
*F. moniliforme*	M/B	x				x								[137]
*F. moniliforme*		x						G			x			[120]
*F. moniliforme*										x	x			[119]
*F. moniliforme*	16S rRNA	x												[89]
*F. moniliforme*	M/16S rRNA				x	x								[138]
*F. moniliforme*	M/B	x			x	x								[96]
*F. moniliforme*	M/16S rRNA	x			x	x								[93]
*F. moniliforme*	B	x				x								[94]
*F. moniliforme*	M/B/16S rRNA	x			x	x								[110]
*F. moniliforme*		x			x	x								[111]
*F. moniliforme; F. oxysporum*	M/B/16S rRNA	x			x	x				x	x			[139]
*F. moniliforme; F. oxysporum; F. semitectum*	M/B	x			x	x								[98]
*F. moniliforme; F. oxysporum; F. semitectum; F. solani*	M/B/16S rRNA	x			x	x				x				[97]
*F. oxysporum *; F. solani**		x											x	[123]
*F. poae*	M/B	x			x	x								[140]
*F. poae*	M/ 16S rRNA	x												[141]
*F. poae,* *F. avenaceum, F. culmorum*		x												[142]
*F. proliferatum*	M/B/16S rRNA	x				x								[143]
*F. subglutinans; F. sambucinum*		x												[144]
*F. verticillioides*	16S rRNA	x			x	x								[92]
*F. verticillioides*		x			x	x								[104]
*F. verticillioides*	16S rRNA	x			x	x							x	[121]
*F. verticillioides*		x		x									x	[122]
*F. verticillioides*	16S rRNA	x			x	x								[145]
*F. verticillioides; F. oxysporum*	M/B/16S rRNA	x			x	x								[90]
*Fusarium graminearum, F. culmorum, F. oxysporum, F. sporotrichiella, F. moniliforme*		x				x		F	x	x	x			[146]

**Table 2 ijms-20-06119-t002:** Patent lists of *Streptomyces* spp. (and derived molecules) against *Fusarium* toxigenic species. Data were obtained combining the results of Espacenet and Orbit Intelligence with the following search words, “*Fusarium*” and “*Streptomyces*”, limiting the period of publication from 2000 to 2018.

Publication Number	Publication Date	Target *Fusarium* spp.	Source	Reference
RU2003100579 A	27/07/2004	*F. moniliforme, F. sambucinum, F. avenaceum*	Espacenet	[150]
KR100914225 B1	26/08/2009	*F. graminearum*	Espacenet	[151]
CN101698827 B; CN101698827 A	28/04/2010	*F. moniliforme*	Espacenet	[152]
CN101822272 A	8/09/2010	*F. avenaceum, F. semitectum*	Orbit Intelligence	[153]
KR101098280	23/12/2011	*F. proliferatum*	Orbit Intelligence	[154]
CN102433281 A; CN102433281 B	2/05/2012	*F. graminearum*	Espacenet	[155]
KR101211681	12/12/2012	*F. fujikuroi*	Orbit Intelligence	[156]
CN102835423 B; CN102835423 A	26/12/2012	*F. nivale, F. graminearum*	Espacenet	[157]
CN103114064 B; CN103114064 A	22/05/2013	*F. moniliforme, F. graminearum*	Espacenet	[158]
CN103820351 A; CN103820351 B	28/05/2014	*F. moniliforme, F. graminearum*	Espacenet	[148]
CN104130965 A	5/11/2014	*F. moniliforme*	Espacenet	[159]
CN104140982 A	12/11/2014	*F. moniliforme*	Espacenet	[160]
CN105060951 A	18/11/2015	*F. moniliforme*	Espacenet	[161]
EP3048890 A1	3/08/2016	*F. culmorum*	Orbit Intelligence	[162]
CN105886428 A	24/08/2016	*F. verticillioides*	Espacenet	[163]
CN106676040	17/05/2017	*F. graminearum*	Orbit Intelligence	[164]
CN107058131	18/08/2017	*F. graminearum*	Orbit Intelligence	[147]
CN107164259 A	15/09/2017	*F. culmorum*	Espacenet	[165]
CN107287130 A	24/10/2017	*F. verticillioides*	Espacenet	[166]
WO201553482 A1	16/04/2018	*F. proliferatum*	Orbit Intelligence	[167]
CN108048380 A	18/05/2018	*F. graminearum*	Espacenet	[168]
CN108102961 A	1/06/2018	*F. graminearum*	Espacenet	[169]
CN108165506	15/06/2018	*F. graminearum*	Orbit Intelligence	[170]
CN108208016	29/06/2018	*F. graminearum*	Orbit Intelligence	[149]
CN108587981	28/09/2018	*F. graminearum*	Orbit Intelligence	[171]

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
