# Peer review of "Critical Assessment of Streptomyces spp. Able to Control Toxigenic Fusaria in Cereals: A Literature and Patent Review"

_ijms, 2019, doi:10.3390/ijms20246119_

Round 1

Reviewer 1 Report

The manuscript is a very interesting review concerning the use of Streptomyces species to control Fusarium contaminations and their mycotoxins in grains. It is a rigorous work that made an extensive and comprehensive coverage of current state-of-the-art in this topic. I thoroughly recommend the publication of the manuscript. I attach the document with minor corrections to be done by the authors. They mostly concern to references. I also have one question. Since Streptomyces species are a source of antibiotics (for bacteria), there is or not a risk of disseminating some of those strains in the fields? I think it is a topic that could be developed a lite beat by the authors in this manuscript. I also recommend to authors to make an effort to reduce the size of table 1. The way it is, it is quite challenging to do a proper reading. Maybe inverting the orientation of the table, or using single space lines, or leaning the table headers.

Author Response

We would like to acknowledge the reviewer for the suggestions as they will improve the quality of the manuscript. We indeed addressed all the suggested corrections. We also tried to re-size Table 1: we summarized the column addressing the study of mechanisms of action and abbreviated the environments selected for in vivo validations. We couldn’t eliminate some columns even though they are empty, because we wanted to stress the lack in the current research dealing with biocontrol streptomycetes against fusaria. In addition, regarding the question found in revision file, we have found that the majority of the studies were focused on DON producers, but as we explained in the review core, also other mycotoxins have been evaluated. We corrected the mistakes present in the reference list, including the patent numbers. When we didn’t specify the range of page number, it is due to the type of article: online-only articles have only volume and the number of article.

The spread of BCAs or metabolites produced by them in the environment, as pointed out by the reviewer, is a real risk to be taken into account. Therefore, during streptomycete selection, the determination of antibiotics production and their identification, as well as monitoring the spread of these compounds in the environment, has to be performed. This topic (still object of discussion within the scientific community) was object of other review (for example Reference 44 see below):

Rey, T.; Dumas, B. Plenty is no plague: Streptomyces symbiosis with crops. Trends Plant Sci. 2017, 22, 30–37.

Therefore, as we are conscious about the risks of using these antibiotic producing bacteria, we stressed the importance of a precise species and strain definition as well as identification of produced metabolites can contribute to a sort of risk assessment for their use in agriculture during the screening process.

See lines 117-119:

Moreover, from a food safety perspective, Streptomyces isolates are able to produce dangerous metabolites for human and animal health, such as antimycin A found on wheat and barley grains [52]. Therefore, it is essential that species and strain characterization is performed accurately.

And lines 135-137:

Evaluation of the antifungal mode of action. Risks concerning the use of these antibiotic producing bacteria associated to events of horizontal gene transfer and the development of antibiotic resistance are still under debate within the scientific community [44]. However, given the current legislative requests [59], understanding the mode of action is essential in order to proceed with the registration of a BCA, in order to avoid risks of spreading in the environment dangerous metabolites for human and animal health [60].

Reviewer 2 Report

This review manuscript is well writtend and organized, the authors made a wide and deep overview of the literature and patents published on the use of Streptomyces species as biocontrol agents of Fusarium in cereals. Also the vantages and disadvantages, together with the pitfalls were well analyzed and discussed. This review could be used for the research community to have a good scenario and update of the possibel use of Streptomyces in biological control of Fusarium in cereal. The only important in my opinion to be improved is the table 1, that is not easy to read and dispersive, too much columns that often are empty, should be summarized in my opinion. Line 48 page 2 please write directly F. verticillioides (syn F. moniliforme) because is more than 15 years that it has been renamed.

The manuscript is worthy of publication in my opinion after addressing these minor points

Author Response

We thank the reviewer for the comments, because they certainly improve the quality of the manuscript. We tried to re-size Table 1: we summarized the column about the study of mechanisms of action and gave some abbreviations regarding the environment selected for in vivo validations. We couldn’t eliminate some columns although they are empty because we wanted to stress the lack in the current research dealing with biocontrol streptomycetes against fusaria. Moreover, we corrected the line 48: we are conscious that F. moniliforme is an incorrect species classification, however we kept it because in some of the articles which we refer to (although they are really recent) they still use this species name and likely is a result of a misleading classification. Therefore, to be more clear with the reader we kept both nomenclatures.